# Comparative Analysis of the Impact of Three Drying Methods on the Properties of *Citrus reticulata* Blanco cv. Dahongpao Powder and Solid Drinks

**DOI:** 10.3390/foods12132514

**Published:** 2023-06-28

**Authors:** Shunjie Li, Xiaoxue Mao, Long Guo, Zhiqin Zhou

**Affiliations:** 1College of Horticulture and Landscape Architecture, Southwest University, Chongqing 400716, China; 19562102160@163.com (S.L.); mofei770501@163.com (X.M.);; 2Key Laboratory of Horticulture Science for Southern Mountainous Regions, Ministry of Education, Chongqing 400715, China; 3The Southwest Institute of Fruits Nutrition, Banan District, Chongqing 400054, China

**Keywords:** *Citrus reticulata* Blanco cv. Dahongpao, three drying methods, powder, solid drinks

## Abstract

*Citrus reticulata* Blanco cv. Dahongpao is a traditional Chinese citrus variety. Due to the high investment in storage and transport of *Citrus reticulata* Blanco cv. Dahongpao and the lack of market demand, the fresh fruit is wasted. The processing of fresh fruit into fruit drinks can solve the problem of storage and transport difficulties and open up new markets. Investigating the effects of different drying processes (hot air, freeze, and spray drying) on fruit powders is a crucial step in identifying a suitable production process. The experiment measured the effects of different drying methods (hot air drying, freeze drying, and spray drying) on the nutrient, bioactive substance, and physical characteristics of fruit powder. This study measured the influence of three different drying methods (hot air, freeze, and spray drying) on the nutritional, bioactive substance, and physical characteristics of fruit powder. The results showed that compared to vacuum freeze-drying at low temperature (−60 °C) and spray-drying at high temperatures (150 °C), hot air drying at 50 °C produced fruit powder with superior nutritional quality, higher levels of active substances, and better physical properties. Hot air drying produced fruit powder that had the highest content of amino acids (11.48 ± 0.08 mg/g DW), vitamin C (112.09 ± 2.86 μg/g DW), total phenols (14.78 ± 0.30 mg/g GAE DW), total flavonoids (6.45 ± 0.11 mg/g RE DW), organic acids, and antioxidant activity capacity. Additionally, this method yielded the highest amounts of zinc (8.88 ± 0.03 mg/Kg DW) and soluble sugars, low water content, high solubility, and brown coloration of the fruit powder and juice. Therefore, hot air drying is one of the best production methods for producing high-quality fruit powder in factory production.

## 1. Introduction

Citrus, one of the largest fruit crops planted in China, boasts high yield and economic efficiency [1]. However, there is still room for improvement in the development of citrus-related industries. Food processing into powder is a common practice in the food industry. The conversion of citrus into fruit powder can address the issues related to non-storage tolerance, vulnerability, and transportation inconvenience while broadening the prospects for extensive application of citrus powder in seasoning, auxiliary additives, solid beverages, and other potential domains [2]. The processing technique of fruit powder is closely related to the retention of nutrients, appearance quality, and physical characteristics of the fruit. This is essential in meeting consumer demands. Choosing an appropriate production process plays a significant role in guiding factory production [3].

“Dahongpao” red tangerine (*C. reticulata* Blanco cv. Dahongpao) is a high-quality citrus variety that has been cultivated in China for 1800 years. However, with the market becoming more diverse and offering a variety of citrus species, *C. reticulata* Blanco cv. Dahongpao have lost some of their appeal due to their sour taste, high seed count, concentrated ripening period, and intolerance to storage [4]. Unfortunately, during the process of discarding them, valuable nutrients and bioactive substances within the red mandarins, such as polymethoxyflavones, are also lost. *C. reticulata* Blanco cv. Dahongpao have a long history of being an important component of traditional Chinese medicine, specifically the rind, which has properties such as regulating energy, invigorating the spleen, drying dampness, and resolving phlegm [5]. Additionally, seeds can be used as biodiesel feedstock [6]. By promoting the development of the industry and benefiting fruit farmers, the deep processing of *C. reticulata* Blanco cv. Dahongpao can increase their commercial value [7].

Fruit and vegetable juices are high in sugars, making the production of solid beverages a challenge as they cannot be dried directly to powder [8]. Therefore, carrier materials such as maltodextrin, gum Arabic, and pectin are added during the drying process. Carrier materials serve to encapsulate bioactive compounds, lower viscosity, and act as drying aids. The encapsulation process is widely used in food, pharmaceutical, nutritional, and cosmetic industries to package and protect active substances from environmental factors like light, air, temperature, and moisture, preserving compound stability [9]. Drying techniques such as spray drying and freeze drying are commonly used for the encapsulation of bioactive compounds, extending shelf life and allowing for controlled release under specific conditions, resulting in better quality products [10].

Spray drying is a simple and cost-effective method to produce dry granular or powder. It involves four steps: preparing the liquid with carrier material, homogenization, atomization, and dehydration [11]. However, high temperatures may cause nutrient loss. By modifying process variables and adding carrier materials, this problem can be reduced [12]. Handling citrus and mulberries is difficult due to their sugar and organic acid content, but carrier materials can help improve the process [13].

Freeze drying is a process that removes water from frozen solutions through sublimation at low temperature and pressure [14]. It involves four steps: preparing the dispersion medium, freezing, sublimation, and evaporation [15]. Freeze drying preserves nutrients, color, flavor, and texture in powdered products, including vitamin C and volatile substances [16]. However, it is less efficient and six times more expensive than spray drying, with long production cycles and high instrument costs causing challenges for large-scale industrial production and scale-up applications [13].

Hot air drying exposes a sample to hot air, causing moisture on the surface to evaporate [17]. However, if prolonged, it can decrease quality and volume [18]. Although easy to operate, it is four to eight times more expensive than freeze drying and leads to poor-quality rehydrated products due to high temperatures that degrade heat-sensitive ingredients like bioactive and volatile components [19]. Additionally, it alters the proportion of volatile organic compounds with fruit aroma.

The main aim of this study was to compare the effects of various drying methods on the quality of *C. reticulata* Blanco cv. Dahongpao powder and to identify the optimal drying process for large-scale production. Fruit powder properties, active substances, and functions can change during the drying process; therefore, this study focused on using *C. reticulata* Blanco cv. Dahongpao as the raw material for producing fruit powder. The objectives were (1) to analyze the effects of different drying methods on the amino acid profile, volatile substances, and chemical antioxidant activity of *C. reticulata* Blanco cv. Dahongpao powder; (2) to evaluate the physicochemical properties of the *C. reticulata* Blanco cv. Dahongpao powder produced by different drying methods; and (3) to conduct color difference and sensory evaluations of solid drinks made from the fruit powder obtained by each drying method.

## 2. Materials and Methods

### 2.1. Citrus Material

*Citrus. reticulata* Blanco cv. Dahongpao was harvested from Puya Village, Dazhou Town, Wanzhou District, Chongqing, and transported to the Key Laboratory of agricultural biosecurity and green production, ministry of Education, School of Horticulture and landscape architecture, southwest university (Chongqing, China). The fruits were selected to be in good shape and free from pests and diseases; they were washed and dried, and then placed in a −20 °C refrigerator for freezing until use.

### 2.2. Fruit Powder Obtained by Different Drying Methods

The juice was obtained by homogenizing the *Citrus. reticulata* Blanco cv. Dahongpao samples in a 200-mesh gauze using a wall breaker. The fruit pomace was dried and then crushed before being filtered through a 60-mesh sieve. The resulting pectic acid extract was heated in a water bath with a citric acid solution at pH 2.0 for 2 h at 95 °C, followed by filtration and cooling to room temperature. Pectin was washed 2–3 times using anhydrous ethanol, and then dried at 40 °C in an oven. Finally, pectin powder was obtained by grinding it with a grinder and set aside. The filtered juice was subjected to three drying methods: freeze drying, spray drying, and hot air drying. Before drying, 200 g of the filtrated juice was pretreated by adding 3 g/Kg of *Citrus. reticulata* Blanco cv. Dahongpao pectin and 10 g/Kg of β-maltodextrin by mass of the juice, heating it at 60 °C, stirring well until dissolved, filtering through a 200-mesh gauze, and allowing the filtrate to dry. For freeze drying, the samples were frozen at −80 °C for 8 h and then dried in a freeze dryer at −60 °C under vacuum at a pressure of less than 20 Pa for 24 h. The dried samples were homogenized by grinding with a rod. For spray drying, the sample was fed into the spray dryer using a peristaltic pump. The drying inlet air temperature was 150 °C, the outlet air temperature was 65–75 °C, the fan was set at 60 Hz, the peristaltic pump was 400 mL/h, and the needle was passed for 2 s. The drying was carried out for 15 min, after which the samples were homogenized by grinding with a rod. For hot air drying, the samples were placed on trays lined with tin foil and greaseproof paper and dried for 72 h in a blast oven at 50 °C. The dried samples were homogenized by grinding with an abrasive rod. All dried and ground samples were packed in sealed bags and stored in a refrigerator at −20 °C for testing.

### 2.3. Analysis of Nutrient Content

Soluble sugar: The soluble sugar content was determined by referring to the national standard GB 5009.8-2016 “Determination of fructose, glucose, sucrose, maltose and lactose in foodstuffs” with slight modifications. The samples were extracted from 1 mL (liquid) or 0.25 g (solid) in a beaker and fixed with distilled water to 80 mL. The samples were sonicated for 10 min at room temperature and then centrifuged at 4000 rpm for 5 min. The supernatant was filtered through a 0.22 μm aqueous membrane and then detected by HPLC. Organic acid: The organic acid content was detected by referring to the method of Yuan Zhou et al. with slight modifications [20]. The samples were extracted from 2 mL (liquid) or 0.25 g (solid) in a 200 mL beaker, and the volume was fixed to 160 mL by adding distilled water. The samples were sonicated at room temperature for 10 min and then centrifuged at 4000 rpm for 5 min. The resulting supernatant was filtered through a 0.22 μm aqueous membrane and analyzed by HPLC. Mineral element: Mineral element contents (Ca, Zn, Fe, Cu, Mg, K, Mn, Na) were determined using 10 mL (for liquid samples) or 1 g (for solid samples) of the sample in a conical flask with 30 mL of mixed acid (HNO_3_:HClO_4_ = 4:1), soaked overnight with a small funnel at the mouth [21]. Digestion was performed on an adjustable hot plate at 120 °C for 2–3 h until the remaining solution amounted to 1–2 mL. After adding distilled water to make up the volume of 10 mL, FAAS was used to determine K, Ca, Na, Fe, and Zn, while GFAAS was used to determine Cu and Mn. For P determination, 1 g (for solid samples) or 1 mL (for liquid samples) of the sample was weighed in a conical flask with 10 mL of HNO_3_, 1 mL of HClO_4_, and 2 mL of H_2_SO_4_. After digestion on an adjustable electric plate and subsequent addition of distilled water to fix the volume at 10 mL, P content was determined using molybdenum blue spectrophotometry with slight modifications based on national standard GB 5009.87-2016

### 2.4. Analysis of the Amino Acid Content 

A total of 2.5 mL (liquid) or 1.25 g (solid) of a sample in an amino acid hydrolysis tube was weighed, 20 mL of a 6 mol/L HCl and 1 mL of 1% phenol was added and shaken well. The vacuum pump was evacuated until there were no air bubbles and then filled with nitrogen until all air was removed and the cap was tightened. The solution was then hydrolyzed in an oven at 110 °C for 24 h. The solution was left at room temperature, cooled, and centrifuged at 4000 rpm for 10 min. The supernatant was spun at 80 °C until nearly dry; 2 mL of distilled water was added, spun again, and repeated twice for a final volume of 6 mL. A total of 1 mL of the volume solution was taken into a test tube together with 500 μL of a phenyl isothiocyanate–acetonitrile solution (0.1 mol/L) and 500 μL of a triethylamine–acetonitrile solution (0.1 mol/L). The solution was mixed well and placed in a water bath at 50 °C for 45 min; then, 2 mL of n-hexane was added. It was then vortexed for 1 min and left to stand. The solution was allowed to delaminate and pass the lower layer through a 0.45 μm organic filter membrane for the determination of amino acids using HPLC.

### 2.5. Analysis of Bioactive Substances

Total phenolic: The total phenolic content of the *C. reticulata* Blanco cv. Dahongpao extract was determined by the Folin–Ciocalteu method described by Singleton et al. with slight modifications [22]. A total of 120 μL of extract was added to 4 mL of distilled water and 400 μL of the Folin–Ciocalteu reagent. After 5 min in the dark, 2 mL of Na_2_CO_3_ solution (5%) was added. After 60 min of reaction in the dark, the total phenolic content was measured by spectrophotometer at 765 nm. Total flavonoid: Total flavonoid content was determined by the NaNO_2_-Al(NO_3_)_3_ colorimetric method, i.e., 200 μL of a 5% NaNO_2_ and 700 μL of distilled water were added to the sample and reacted for 6 min, followed by 300 μL of a 10% Al(NO_3_)_3_ and reacted for another 6 min. Finally, 2 mL of NaOH solution (1 mol/L) was added; the volume was made up to 5 mL with distilled water, and the total flavonoid content was measured by spectrophotometer at 500 nm after 15 min of reaction. Carotenoid: Carotenoid content was measured by the method of Delphine Amah et al. with some adjustments [23]. Samples were extracted using an extraction solution of hexane:acetone:ethanol (2:1:1) containing 0.1% BHT at a ratio of 1:5 (solid: liquid). Extraction was carried out by ultrasonication for 40 min at room temperature and repeated twice, with supernatants combined thereafter. The extract was centrifuged at 6500 rpm for 5 min at 4 °C, and the organic phase was collected after standing and layering. The collected liquid was spin-distilled to near dryness at 40 °C, re-dissolved with MTBE (with 0.1% BHT), and fixed to 2 mL. Saponification was carried out for 12 h by adding 2 mL of a 10% methanol–KOH solution (with 0.1% BHT). Then, 5 mL of distilled water and 2 mL of MTBE were added for extraction. The organic phase was collected, and the extraction was repeated twice with a 2 mL MTBE. The organic phase was combined and dried using 0.25 g of anhydrous Na_2_SO_4_ for 30 min to 1 h. The organic phase was poured out and nitrogen-blown to dryness, then re-dissolved with MTBE and fixed to 1.5 mL before being filtered through a 0.22 μm organic phase filter membrane and analyzed by HPLC. Vitamin C: The vitamin C content was measured following the national standard GB 5009.86-2016 “National standard for food safety Determination of ascorbic acid in food,” with slight adjustments. Samples were extracted through ultrasonication at room temperature for 5 min, centrifuged at 4000 rpm for 5 min, and the resulting supernatant was filtered through a 0.45 μm aqueous phase membrane and analyzed by HPLC.

### 2.6. Analysis of Volatile Substances

A solid sample of 1 g was weighed in a 20 mL screw-top headspace flask; then, 5 mL of saturated NaCl solution and 20 μL of hexane (containing 263.6 μg of cyclohexanone) was added. The cap was sealed with a PTFE spacer, quickly screwed down and mixed, and detected using GC-MS. The volatiles were extracted using headspace solid-phase micrography. We characterized the volatiles by comparing their constituent components to those listed in the NIST Chemistry WebBook spectral library (NIST Standard Reference Database No. 69) and selecting only those matches with a similarity score greater than 80%.

### 2.7. Analysis of the Chemical Antioxidant Activity

In this experiment, three methods were used to determine the antioxidant activity of *C. reticulata* Blanco cv. Dahongpao fruit powder under different drying methods: ABTS radical scavenging ability, DPPH radical scavenging ability, and FRAP iron ion reduction ability [24]. An 80% methanol-dissolved Troloxs reagent was used to make standard curves, and the results are all expressed as TroloxEquivalent (TE) per mL of fresh weight (μmol/mL TE FW). All samples were diluted 5 times with distilled water and left to be measured. For ABTS analysis, 5 mL of an ABTS aqueous solution (7 mmol/L) was mixed with 88 μL of a K_2_S_2_O_4_ solution (140 mmol/L) and protected from light for 12–16 h. The ABTS reaction solution was diluted with anhydrous ethanol to an absorbance value of 0.70 ± 0.02 (734 nm) before use. A total of 40 μL of the sample or Trolox solution was reacted with 3.90 mL of an ABTS reaction solution for 10 min under light-proof conditions and then measured at 734 nm. For DPPH analysis, 100 μL of the sample or Trolox solution was added to 3.80 mL of the DPPH solution (75 μmol/L), protected from light for 30 min, and then measured at 517 nm. For FRAP analysis, acetate buffer (0.3 mol/L, pH = 3.6), a FeCl_3_ solution (20 mmol/L), and a TPTZ working solution (10 mmol/L) were prepared using HCl (40 mmol/L), and the above three solutions were mixed in a certain ratio (10:1:1) to obtain the TPTZ solution. The absorbance was measured at 593 nm after the reaction of 100 μL of the sample or Trolox solution with 3.90 mL of the TPTZ solution for 30 min under light-proof conditions.

### 2.8. Analysis of the Physicochemical Properties

The solubility and oil absorption were determined via the method of Yoong–Kong How et al. [25]. The solubility was determined by weighing 1.0 g of fruit powder in a 50 mL centrifuge tube and adding 40 mL of distilled water, then placed in a magnetic stirrer for 5 min and centrifuged at 4000 rpm for 5 min. A total of 10 mL of the resulting supernatant was placed in a glass Petri dish and dried at 105 °C for 3 h. The sample was removed and cooled in a desiccator and the total mass was weighed. The solubility was calculated as follows:(1)X=[(m1−m2)×V1×100]/[m3×V2−(1−w)],
where *X* represents the solubility of the sample, %; *m*1 represents the total mass of the solid residue and the Petri dish after drying, g; *m*2 represents the mass of the Petri dish after drying, g; m3 represents the mass of the fruit powder, g; *V*1 represents the volume of distilled water added, 40 mL; *V*2 represents the volume of the supernatant taken, 10 mL; *w* represents the water content of the fruit powder, %.

The oil absorption was determined by adding 5 mL of rapeseed oil to a 10 mL centrifuge tube. A total of 1.0 g of fruit powder was slowly added to the tube and mixed by vortex shaking while adding. It was allowed to stand for 30 min and centrifuged at 5000 rpm for 20 min; then, the volume of the upper oil layer was measured. The oil absorbency was calculated using the following formula:(2)X=(V1−V2)/m,
where *V*1 represents the volume of rapeseed oil added, mL; *V*2 represents the volume of the top oil layer after centrifugation, mL; *m* represents the mass of fruit powder, g.

Fruit powder yield (%) = mass of fruit powder/mass of dried sample to calculate the yield of *C. reticulata* Blanco cv. Dahongpao fruit powder for different drying methods.

### 2.9. Analysis of the Color Difference of Citrus. reticulata Blanco cv. Dahongpao Solid Drinks under Different Drying Methods

A total of 5 g of fruit powder was weighed in a centrifuge tube, distilled water was added to a volume of 10 mL and set aside for measurement. The sample (solid/liquid) was stirred well and six spots were taken with a white paper background. The values of L* (brightness), a* (red–green deviation), b* (blue–yellow deviation), C* (saturation), and h° (hue angle) were measured and recorded using a MINOLTA CR-300 (D65 light source) colorimeter. The color difference composite index was calculated according to the method of Sdiri et al. (2012) with the following equation:(3)$$CCI=1000a^{*}/\left(L^{*}\times b^{*}\right),$$
where L*, a*, b* represents the lightness index and color index of the sample in the Lab system.

### 2.10. Sensory Evaluation of C. reticulata Blanco cv. Dahongpao Solid Drinks under Different Drying Methods

The sensory evaluation was based on the method of Min Zheng et al. [21], and the weighted scoring method was used to determine the percentage of scores for different sensory characteristics by weighting coefficients to make a comprehensive evaluation of the fruit powders with slight modifications. The brewing was carried out according to the recommended brewing ratio and method for solid drinks in the market, specifically the fruit powder was brewed in a transparent container with warm water (80 ± 1) °C at a ratio of 1:60 (g/mL), pending evaluation. A 10-member evaluation team was set up to evaluate the 4 indicators of color, taste, odor, and stability of the fruit powder. We calculated the weight of each indicator using the forced decision method, as described in the literature. To obtain the final trial score, we multiplied the score of each indicator by its corresponding weight, summed these products, and then divided by the number of evaluators. The specific sensory evaluation table (Table 1) is shown below.

### 2.11. Statistical Analysis

All data collected were processed using Excel 2018, analyzed, and plotted using Prism 9.0, and significant differences between samples were assessed by one-way ANOVA and Duncan’s multiple tests, with differences considered significant when *p* < 0.05. Pearson correlation analysis was performed using Origin Pro 2021 software. All experiments were repeated three times and data results were expressed as mean ± standard deviation (SD).

## 3. Results

### 3.1. Analysis of Nutrient Content 

Figure 1A shows the organic acid content of the *C. reticulata* Blanco cv. Dahongpao powder obtained from the three drying processes. None of the dried fruit powders contained oxalic acid. The tartaric and acetic acids were found to be present in the highest amount in the spray drying process, with a concentration of 30.56 ± 0.83 mg/g DW and 129.34 ± 0.14 mg/g DW, respectively. On the other hand, the other four acids had the highest concentration in the fruit powder obtained by hot air drying. In this case, the hot-air-dried fruit powder had the lowest content of acetic acid due to decarboxylation during the drying process [26].

Figure 1B shows that only three soluble sugars were found in *C. reticulata* Blanco cv. Dahongpao fruit powder obtained by spray drying, and sucrose was not detected in either freeze drying or hot air drying. Glucose had the highest content of soluble sugars in *C. reticulata* Blanco cv. Dahongpao powder obtained through all three drying processes, ranging from 117.27 mg/g DW to 155.23 mg/g DW. On the other hand, sucrose had the lowest content of soluble sugars. Freeze drying produced a higher glucose content, while hot air drying resulted in a higher fructose content.

Figure 1C presents the mineral element contents of *C. reticulata* Blanco cv. Dahongpao fruit powder obtained from three drying processes. Ca, Mg, P, and K did not differ significantly among the five major elements, ranging from 0.87 mg/Kg DW to 16.11 mg/Kg DW. Na content differed significantly, with freeze-dried fruit powder having the highest at (20.35 ± 1.00) g/Kg DW and spray-dried fruit powder having the lowest at (17.29 ± 0.69) g/Kg DW. Fe, Cu, and Mg contents for the four trace elements were insignificant, ranging from 0.35 mg/Kg DW to 1.34 mg/Kg DW. However, Zn content showed a significant difference, with hot-air-dried fruit powder having the highest Zn content at 8.88 ± 0.88 mg/Kg DW, while freeze drying produced fruit powder with the lowest Zn content at 4.83 ± 0.32 mg/Kg DW, with a difference of 4.05 mg/Kg DW between the two methods.

### 3.2. Analysis of the Amino acid Content 

Figure 2 displays the amino acid content and types in *C. reticulata* Blanco cv. Dahongpao powder obtained through different drying methods [27]. Hot air drying resulted in the highest total amino acid content at 11.48 ± 0.08 mg/g DW, followed by freeze drying at 10.07 ± 0.04 mg/g DW, while spray drying resulted in the lowest content at 8.98 ± 0.06 mg/g DW. Isoleucine (Ile) had the highest content in all three drying methods, while proline (Pro) had the lowest content. Freeze drying produced high contents of arginine (Arg), threonine (Thr), and methionine (Met), while spray drying produced high contents of alanine (Ala) and proline (Pro). Hot air drying resulted in the highest contents of other amino acids.

Table 2 shows the variety of amino acids detected in *C. reticulata* Blanco cv. Da-hongpao fruit powder obtained through freeze drying, hot air drying, and spray drying with varying levels. The study found that these drying methods produce fruit powders rich in the same types of amino acids. Hot air drying produces fruit powder with the highest content of both essential and non-essential amino acids compared to other methods, including high concentrations of specific amino acids such as acidic (Glu, Asp), hydrophilic (Ser, Thr), hydrophobic (Gly, Ala, Leu, Pro, Met, Phe, Ile, Trp), aromatic (Phe, Trp), and branched-chain (Ile, Leu) amino acids.

### 3.3. Analysis of Bioactive Substances

As shown in Figure 3A,B. *C. reticulata* Blanco cv. Dahongpao powder obtained through hot air drying had the highest phenol and flavonoid content at 14.78 ± 0.30 mg/g GAE DW and 6.45 ± 0.11 mg/g RE DW, respectively [28]. The gradual rupture of plant cell structure during hot air drying led to the release of phenols from the cell matrix, while the prolonged high temperature protected phenols in the system from enzymatic depletion [29]. Spray drying had lower contents with 10.12 ± 0.80 mg/g GAE DW and 5.31 ± 0.25 mg/g RE DW, respectively. Freeze drying produced the least total phenols and total flavonoids at 8.50 ± 0.73 mg/g GAE DW and 4.65 ± 0.22 mg/g RE DW, respectively).

Figure 3C shows that spray drying produced the highest content of carotenoids, particularly lutein and β-cryptoxanthin, at 131.97 ± 2.58 μg/g DW and 130.45 ± 1.81 μg/g DW, respectively. Hot air drying resulted in higher lutein content but lower levels of the other three carotenoids compared to freeze drying.

Figure 3D shows that hot air drying produced the highest vitamin C content at 112.09 ± 2.86 μg/g DW, followed by freeze drying at 101.42 ± 1.09 μg/g DW, and then spray drying with the lowest content at 22.24 ± 0.58 μg/g DW. Hot air drying resulted in more vitamin C retained due to temperature changes and airflow, while spray drying destabilized vitamin C at high temperatures, leading to significant losses during processing. Freeze drying, carried out under low-temperature vacuum conditions, reduced vitamin C degradation.

### 3.4. Analysis of Volatile Substances 

The study analyzed the volatile substances in three types of *C. reticulata* Blanco cv. Dahongpao powder produced by different drying methods. As shown in Table 3, The results showed that freeze drying generated *C. reticulata* Blanco cv. Dahongpao powder with the most abundant volatile substances, including 19 types such as monoterpenes, oxygen-containing compounds, alkanes, esters, and ketones. In contrast, hot air drying and spray drying had a similar number of volatile substances, both consisting of 14 types. This may be due to the low temperature and vacuum ability to retain volatile organic compounds, while high temperature and prolonged heating can cause the decomposition of these organic compounds, altering their composition and concentration [30]. Among the three types of *C. reticulata* Blanco cv. Dahongpao powder, D-limonene was found to have the highest content of volatile substances. Interestingly, ester and ketone volatile substances were not detected in the hot air-dried fruit powder, while ester volatile substances were not detected in the spray-dried fruit powder. These findings suggest that the freeze drying method is crucial for retaining the volatile organic compounds, which contributes to the flavor and aroma of *C. reticulata* Blanco cv. Dahongpao powder.

### 3.5. Analysis of Chemical Antioxidant Capacity 

The antioxidant capacity of *C. reticulata* Blanco cv. Dahongpao fruit powder subjected to different drying processes was evaluated using three methods: ABTS, DPPH, and FRAP. ABTS and DPPH are measurements of radical scavenging ability reflecting the sample’s capacity to eliminate ABTS+ and DPPH+ radicals, while FRAP assesses the reduction capability by measuring the Fe^3+^ to Fe^2+^ reduction capacity. All three evaluation techniques presented varying degrees of improvement. The highest antioxidant capacity was observed in *C. reticulata* Blanco cv. Dahongpao powder obtained via hot air drying, followed by spray drying, and the lowest capacity was observed in samples produced via freeze drying. A thorough analysis of antioxidant capacity across all treatments showed the following: hot air drying method > spray drying method > freeze drying method. The results are shown in Table 4.

### 3.6. Analysis of Compounds and Antioxidant Correlation 

According to Figure 4, all three methods used to determine antioxidant capacity exhibited a positive correlation with the eight indices. The correlation coefficients were 0.9942 and 0.9966 for total flavonoids and antioxidant activity in the ABTS and DPPH assays, respectively, and 0.8689 for total phenols and antioxidant activity in the FRAP assay. These results suggest that total phenols and total flavonoids are more significantly involved in the antioxidant capacity of *C. reticulata* Blanco cv. Dahongpao powder than other substances.

### 3.7. Analysis of Physicochemical Properties

Table 5 shows the results of the tests carried out on the basic physicochemical properties of *C. reticulata* Blanco cv. Dahongpao powder obtained by different drying methods. Hot air drying produced the highest yield at 15.85 ± 0.65%, followed by freeze-drying, while spray drying produced the lowest yield at 11.44 ± 0.14%. Hot air drying resulted in the lowest water content, followed by spray drying, and freeze drying produced the highest water content, which is consistent with the results of Franceschinis et al. [31]. Among the three drying methods, the solubility was ranked from highest to lowest as follows: hot air drying > spray drying > freeze drying. Spray-dried fruit powders contain the highest oil-holding properties at 1.38 ± 0.10 mL/g, while freeze drying has the lowest at 0.39 ± 0.08 mL/g. These properties can improve the taste of food and extend the shelf life of fruit powder.

### 3.8. Analysis of the Color Difference

Table 6 shows the color differences between the powdered form and brewed form of three types of dried *C. reticulata* Blanco cv. Dahongpao powders. In Table 6, “L*” represents the lightness or darkness of the object, expressed on a scale of 0–100 with black at 0 and white at 100; “a*” represents the red–green color of the object, with positive values indicating red; “b*” represents the yellow–blue color of the object, with positive values indicating yellow; “c*” represents color saturation; “h°” represents hue angle, where 0°, 90°, 180°, and 270° correspond to pure red, pure yellow, pure green and pure blue colors. According to Table 6, spray-dried fruit powder has the highest brightness, followed by freeze-dried fruit powder, and hot air-dried fruit powder is the darkest in powdered form. In brewed form, hot air-dried fruit powder appears even darker, while freeze-dried and spray-dried fruit powders are similar in brightness. By comparing the “a*” and “b*” values, the freeze-dried fruit powder shows a redder tone, whereas spray-dried fruit powder appears more yellow. After brewing, spray-dried fruit powder tends to appear more yellow and red than freeze-dried fruit powder. The “a*” and “b*” values of hot air-dried fruit powder are both lower and exhibit a brownish hue. Both the fruit powder and brewed forms show low saturation for hot air drying, while in contrast, spray drying shows the highest level of saturation. The hot air drying process produces the greatest color difference composite index (CCI) due to the formation of darker color resulting from the browning reaction of sugar during extended heating [32]. The differences between freeze drying and spray drying are not significant.

The color differences of the *C. reticulata* Blanco cv. Dahongpao powders obtained from the three drying methods in both powdered and brewed forms are presented in Figure 5. Freeze-dried fruit powder appears orange–yellow, probably due to the better retention of pigments caused by low-temperature vacuum drying. Hot air drying degrades pigments due to prolonged baking, resulting in brownish–brown colored fruit powder. In contrast, spray drying, although at a higher temperature, has a shorter drying time causing partial destruction of pigments, resulting in yellow-colored fruit powder.

### 3.9. Sensory Evaluation 

Ten evaluators were used to evaluate the color and luster, texture, smell, and stability of *C. reticulata* Blanco cv. Dahongpao powder obtained from three drying methods in a ratio of 1:60 using hot water at 80 ± 1 °C. The results are presented in Table 7. From the images of solid drinks in Figure 1, it is clear that the color ratings for all three types of *C. reticulata* Blanco cv. Dahongpao powders were low. The brewed drinks showed a darker and more intense color, which did not align with the typical “transparent” characteristic of regular drinks. The lowest color rating was assigned to hot-air-dried fruit powder, owing to the brownish color resulting from pigment destruction during prolonged baking. For taste, judges rated it as sour and bitter, whereas hot air drying scored relatively high due to the charred flavor from the long baking time masking the inherent sourness and bitterness of the *C. reticulata* Blanco cv. Dahongpao powder. However, the difference in flavor ratings was less pronounced. In terms of stability evaluation, hot air drying was rated the lowest, while freeze drying was rated higher because the *C. reticulata* Blanco cv. Dahongpao powder obtained via hot air drying would settle faster after brewing. Using the weighting of each indicator, we calculated the overall score of sensory evaluation for the *C. reticulata* Blanco cv. Dahongpao powder obtained from the three drying processes, whereby freeze drying > spray drying > hot air drying.

## 4. Discussion

Solid beverages are currently popular in the market due to their small size, portability, ease of transport, and resistance to spoilage. The traditional drying methods used in the production of solid beverages include freeze drying, spray drying, and hot air drying. Freeze drying is considered one of the most effective drying methods for preserving heat-sensitive substances because it utilizes low temperature and low pressure [33]. In this study, the improved processing conditions afforded by freeze drying were observed to result in higher levels of soluble sugars and volatile compounds in the fruit powders. This may be attributed to the gentle drying process that freeze drying employs. In contrast, hot air drying and spray drying led to the thermal decomposition of volatiles due to higher temperatures involved. The high content of low molecular sugars and organic acids can pose difficulties during drying, such as sticky wall problems, which can lead to reduced yields and handling issues [34]. Thus, to overcome these issues, a carrier material was added to the process, with *C. reticulata* Blanco cv. Dahongpao pomace used for pectin extraction to reduce waste of raw materials throughout the entire process. Additionally, β-maltodextrin was added to reduce stickiness. These findings shed light on the importance of selecting the right drying method and optimizing processing conditions for producing high-quality fruit powders suitable for use in solid beverage production.

Spray drying is a popular choice due to its low operating costs and flexibility compared to freeze drying [35]. However, the very high temperature employed in spray drying can result in greater losses. samples For example, Kong et al. (2011) reported a decline in vitamin C content due to thermal decomposition, which was observed when samples were produced via spray drying. In this study, the vitamin C content of the spray-dried samples was notably lower than that of samples produced via freeze drying and hot air drying [11,36]. Furthermore, Luo concluded that intermittent changes in temperature or air circulation help enhance vitamin C retention when fruits are exposed to them [37]. Interestingly, our study found that hot air drying resulted in the highest vitamin C content of all three methods, likely due to the mild drying conditions employed. Hot air drying also produced fruit powder without any browning reaction, while prolonged oven drying in other methods caused carotenoid loss through decomposition, changing the color from orange to brown [38]. While spray drying requires less time and typically retains sensitive substances such as color and nutrition, our study found that it resulted in the highest carotenoid content, likely due to the short drying time and protective effect of the carrier material. Few studies have explored the impact of various drying methods on mineral elements, and our findings indicate that drying has little effect on the content of seven mineral elements except for sodium and zinc [39].

This study found that certain amino acid contents were lower after spray drying at 150 °C, suggesting a possible reaction with glucose [40]. However, it has been discovered that free amino acids generated by proteins can exceed the amount lost due to Maillard reaction when temperature ranges between 40 °C and 60 °C, leading to a higher final free amino acid content [41]. Moreover, Liu et al. noted that Asp content increased considerably during heating and drying [30]. Hot air drying utilizes lower temperatures compared to spray drying, thus preserving the amino acid content while maintaining other beneficial properties of fruit powders. In contrast, freeze drying partially prevents protein degradation to free amino acids through its sublimation process [42]. Nevertheless, pore formation in the particles caused by water sublimation during the freeze drying process can lead to the degradation of bioactive substances, resulting in lower freeze-dried amino acid content, total phenol, and total flavonoid content. To conclude, these findings highlight the advantages of hot air drying for preserving amino acid content and other beneficial properties of fruit powders [43].

The water content of freeze-dried fruit powders was observed to be higher than those of spray-dried ones in various studies [31,44]. Nevertheless, all three drying methods produced fruit powders with a moisture content greater than 7%, which does not meet the requirements for moisture content of solid beverages in the national standard GB/T 29602-2013 Solid Beverages. The solution state of the fruit powders obtained from all three drying methods was turbid after brewing, but this is desirable in fruit juices like citrus, white grape juice, and apple juice that require clarification through filtration and centrifugation [34]. While freeze drying retains aroma and flavor better, resulting in the best sensory evaluation of fruit powders [33], hot air drying preserves amino acid content and other beneficial properties of fruit powders while maintaining their quality.

This study provides reliable guidance for processing and developing *C. reticulata* Blanco cv. Dahongpao. It explores the ways in which different drying methods affect their nutritional composition, taste quality, physical and chemical properties, providing valuable information for their application. The study also offers ideas for improving the production process and product quality of *C. reticulata* Blanco cv. Dahongpao and serves as a reference for future research on similar materials. Although three different drying methods were compared to produce fruit powders, the issue of nutrient loss at high temperatures still requires attention. Additionally, as a food product, its shelf life, the use of preservatives, costs calculation, microorganism testing, counting of active lactic acid bacteria, and basic food indicators such as protein, capacity, fat, and dietary fiber have not yet been evaluated.

## 5. Conclusions

In conclusion, hot air drying, spray drying and freeze drying have their respective advantages and disadvantages. Hot air drying produced fruit powder with the highest content of amino acids, vitamin C, total phenols, total flavonoids, organic acids, antioxidant activity capacity, Zn yield, carotenoids, and soluble sugars. It had low water content, high solubility, and a brown color for both fruit powder and juice. On the other hand, spray-dried fruit powder had the highest mineral elements and soluble sugar content but lower total phenol and total flavonoid content, resulting in the lowest antioxidant activity capacity. Freeze-dried fruit powder had a higher carotenoid content but lower mineral elements, amino acids, and vitamin C content. It had high water content, low solubility, low oil absorption, bright orange color, stable texture, and good aroma and flavor when brewed. Based on its high nutritional value and antioxidant activity, the hot air drying method is preferable for the production of fruit powder. However, further studies can be performed to improve its sensory quality. In summary, three methods have their own characteristics that can be utilized depending on the specific requirements for the product.

## Figures and Tables

**Figure 1 foods-12-02514-f001:**
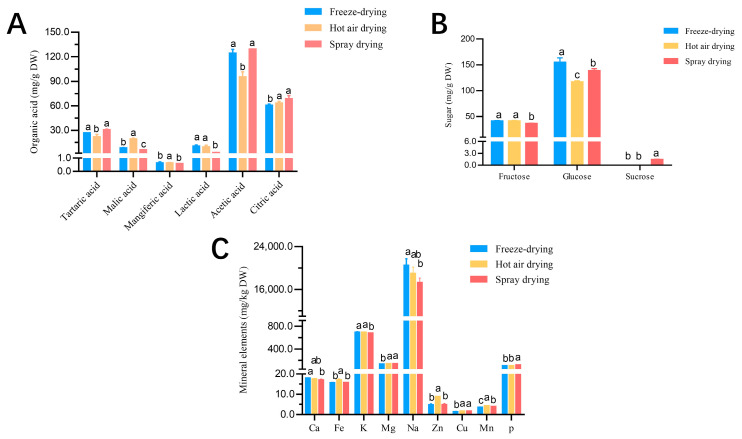
Nutrient content analysis of *C. reticulata* Blanco cv. Dahongpao powder. The content and differences of six organic acids in *Citrus reticulata* Blanco cv. Dahongpao powder produced by three drying methods. (**A**); The content and differences of three soluble sugars in *Citrus reticulata* Blanco cv. Dahongpao powder produced by three drying methods. (**B**); The content and differences of nine mineral elements in Citrus reticulata Blanco cv. Dahongpao Powder produced by three drying methods. (**C**). Note: DW stands for dried weight. Values in the same row with different superscript letters are significantly different (*p* < 0.05).

**Figure 2 foods-12-02514-f002:**
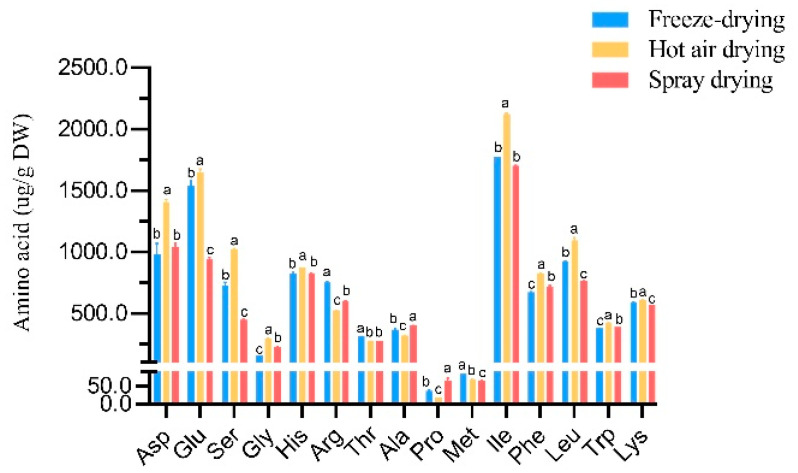
Amino acid content of *C. reticulata* Blanco cv. Dahongpao powder under different drying methods. Fifteen amino acids were detected in *Citrus reticulata* Blanco cv. Dahongpao powder produced by three different drying methods, and differences in the content of each amino acid were analyzed across the different drying methods. Values in the same row with different superscript letters are significantly different (*p* < 0.05).

**Figure 3 foods-12-02514-f003:**
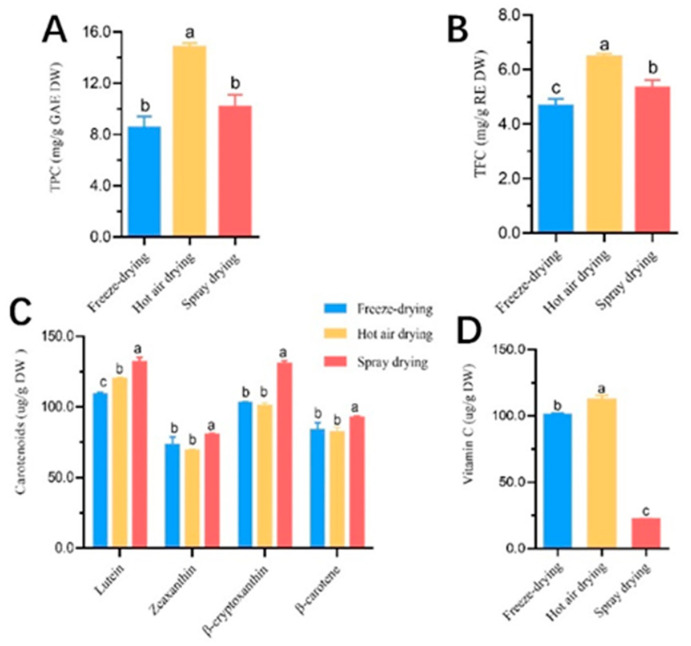
Bioactive substance analysis of *C. reticulata* Blanco cv. Dahongpao powder. Total phenolic content of fruit powder under different drying methods. The content of total phenols differed between the three different drying methods (**A**); total flavonoid content of fruit powder under different drying methods. Differences in the content of total flavonoids under three different drying methods (**B**); Carotenoid species and content of fruit powder under different drying methods. Four carotenoids were detected in the fruit powder produced by three different drying methods, and the content of each carotenoid varied across different species depending on the drying method used. (**C**); and vitamin C content of fruit powder under different drying methods (**D**). Note: GAE stands for Gallic Acid Equivalent; RE stands for Rutin Equivalent; TPC represents total phenolic content; TFC represents the total flavonoid content. Values in the same row with different superscript letters are significantly different (*p* < 0.05).

**Figure 4 foods-12-02514-f004:**
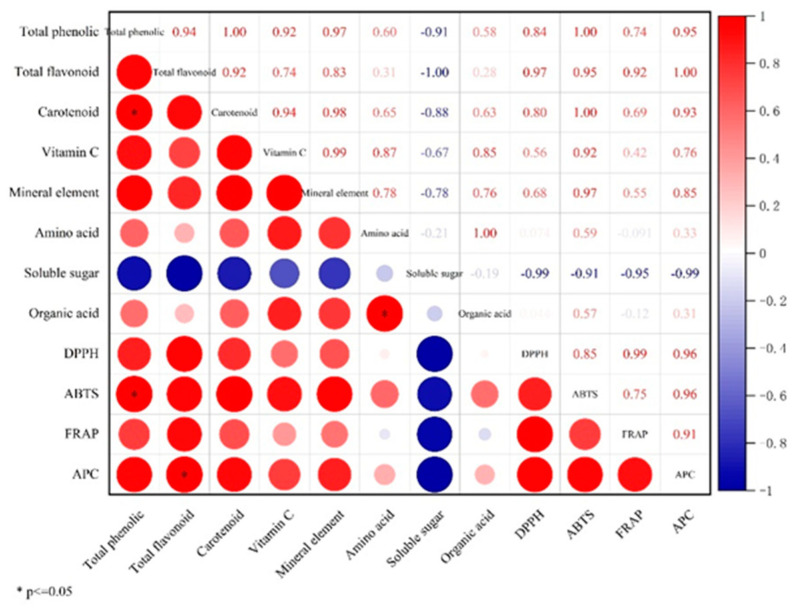
Heat map of the correlation between *C. reticulata* Blanco cv. Dahongpao powder compounds and antioxidant indexes under different drying methods.

**Figure 5 foods-12-02514-f005:**
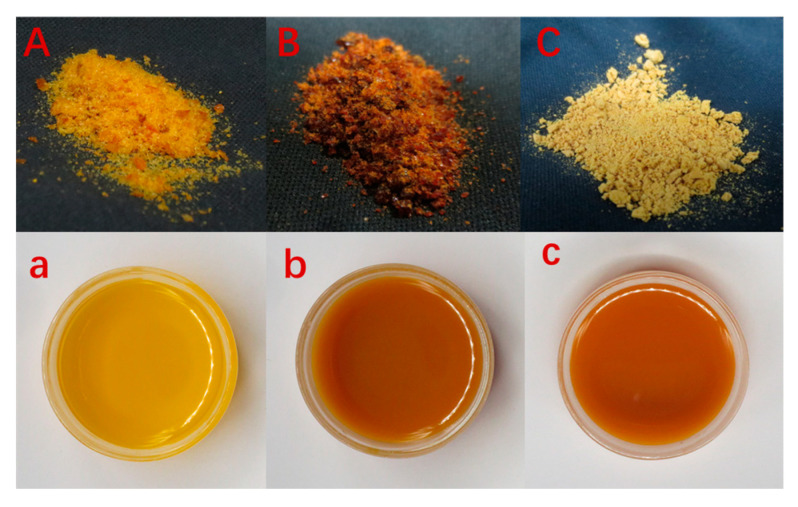
Solid form and brewing form of *C. reticulata* Blanco cv. Dahongpao solid beverage under different drying methods. Note: (**A**,**a**) is freeze drying; (**B**,**b**) is hot air drying; (**C**,**c**) is spray drying.

**Table 1 foods-12-02514-t001:** Sensory Evaluation of Solid Beverage.

Weighting	Excellent (76–100 Points)	Good (51–75 Points)	Medium (26–50 Points)	Poor (0–25 Points)
Color (0.2)	Uniform and translucent, orange in color	Light orange color with a uniform translucency	The color is generally uniform and translucent, yellow	Dull, uneven, and brownish
Taste (0.4)	Light and dense on the palate with a sweet and sour taste of tangerine	The palate is light and dense, with a basic sweet and sour taste	Average taste, sweet or sour	Distinctly offensive odor with a disproportionate sweet and sour taste and a severe bitterness
Scent (0.25)	Aromatic harmony with a citrusy freshness	Average fragrance, with a citrusy note	Inconspicuous citrus aroma	No citrus aroma, bad odor
Stability (0.15)	Clear and transparent, with good stability	Better stability and more homogeneous texture	Average stability and poor textural homogeneity	Visibly cloudy or precipitated

**Table 2 foods-12-02514-t002:** Effect of different drying methods on the amino acid content of *C. reticulata* Blanco cv. Dahongpao powder.

Type	Drying Method		
	Freeze Drying	Hot Air Drying	Spray Drying
Essential amino acids (EAA) mg/g DW	5.53 ± 0.02 ^b^	6.26 ± 0.03 ^a^	5.29 ± 0.01 ^c^
Non-essential amino acids (NEAA) mg/g DW	4.54 ± 0.04 ^b^	5.21 ± 0.06 ^a^	3.70 ± 0.05 ^c^
EAA/NEAA	1.22	1.2	1.43
EAA/TAA	0.55	0.55	0.59
Hydrophilic amino acids (Ser, Thr) mg/g DW	1.03 ± 0.03 ^b^	1.29 ± 0.01 ^a^	0.72 ± 0.01 ^c^
Hydrophobic amino acids mg/g DW (Gly, Ala, Leu, Pro, Met, Phe, Ile, Trp)	4.37 ± 0.02 ^b^	5.14 ± 0.02 ^a^	4.30 ± 0.02 ^b^
Acidic amino acids (Glu, Asp) mg/g DW	2.51 ± 0.08 ^b^	3.05 ± 0.05 ^a^	1.98 ± 0.04 ^c^
Alkaline amino acids (Lys, Arg, His) mg/g DW	2.15 ± 0.02 ^a^	1.99 ± 0.00 ^b^	1.98 ± 0.01 ^b^
Aromatic amino acids (Phe, Trp) mg/g DW	1.04 ± 0.01 ^c^	1.24 ± 0.01 ^a^	1.10 ± 0.01 ^b^
Branched-chain amino acids (Ile, Leu) mg/g DW	2.70 ± 0.00 ^b^	3.21 ± 0.02 ^a^	2.46 ± 0.01 ^c^
Sulfur-containing amino acids (Met) ug/g DW	82.07 ± 0.97 ^a^	67.34 ± 0.62 ^b^	64.20 ± 0.73 ^c^

Note: Values represent means of triplicate determination ± SD. Values in the same row with different superscript letters are significantly different (*p* < 0.05). EAA is Essential amino acids; NEAA is non-essential amino acids; TAA is total amino acids.

**Table 3 foods-12-02514-t003:** Effect of different drying methods on the volatile matter content of *C. reticulata* Blanco cv. Dahongpao powder.

NO.	Compound Name	Drying Method		
		Freeze Drying/%	Hot Air Drying/%	Spray Drying/%
Monoterpenes (8)				
1	α-Pinene	0.22	1.97	2.27
2	β-Pinene	0.71	nd	nd
3	α-Terpinene	0.33	1.6	1.51
4	D-(±)-Limonene	69.8	55.17	46.73
5	α-phellandrene	nd	1.822514	nd
6	p-Mentha-1,4-diene	2.41	nd	nd
7	(1S,3R)-(Z)-4-carene	0.78	2.26	2.36
8	(Z)-β-ocimene	0.56	nd	nd
Oxides (8)				
9	Hexamethylcyclotrisiloxane	1.63	3.77	12.08
10	Decamethylcyclopentasiloxane	2.07	1.98	3.61
11	Dodecamethylcyclohexasiloxane	1.34	1.11	1.8
12	Tetradecamethylcycloheptasiloxane	1.11	1.02	0.94
13	Hexadecamethylcyclooctasiloxane	0.82	0.83	0.85
14	Octadecamethylcyclononyl siloxane	0.74	0.81	0.73
15	Dodecamethylcyclodecasiloxane	0.315	0.26	0.28
16	Octamethylcyclotetrasiloxane	1.7	1.94	12.29
Alkanes (2)				
17	n-Hexane	4.25	24.67	12.06
18	Isooctane	1.06	nd	nd
Esters (1)				
19	n-Butyl chloroacetate	8.4	nd	nd
Ketones (1)				
20	Cyclohexanone	0.76	nd	1.74

**Table 4 foods-12-02514-t004:** Effect of different drying methods on the antioxidant activity of *C. reticulata* Blanco cv. Dahongpao powder.

NO.		Antioxidant Activities (μmol/g TE DW)				
		ABTS	DPPH	FRAP	APC	Rank
1	Freeze drying	192.78 ± 0.85 ^c^	3.05 ± 0.03 ^c^	5.734 ± 0.03 ^c^	52.22	3
2	Hot air drying	352.45 ± 1.47 ^a^	4.00 ± 0.02 ^a^	22.37 ± 0.51 ^a^	100	1
3	Spray drying	279.12 ± 2.01 ^b^	3.47 ± 0.02 ^b^	7.42 ± 0.05 ^b^	66.32	2

Note: Values represent means of triplicate determination ± SD. Values in the same row with different superscript letters are significantly different (*p* < 0.05). ABTS, DPPH, and FRAP represent three different methods for measuring antioxidant activity; APC represents the comprehensive evaluation total score of each antioxidant activity measurement method; rank represents the order of comprehensive antioxidant capacity of each sample; DW stands for dry weight.

**Table 5 foods-12-02514-t005:** Physicochemical properties of *C. reticulata* Blanco cv. Dahongpao solid drinks by different drying methods.

NO.	Drying Method	Yield	Water Content	Solubility	Oil Absorption
		(% DW)	(% DW)	(% DW)	(mL/g DW)
1	Freeze drying	14.18 ± 0.04 ^a^	13.49 ± 0.75 ^a^	77.06 ± 1.30 ^b^	0.39 ± 0.08 ^c^
2	Hot air drying	15.85 ± 0.65 ^a^	9.20 ± 0.46 ^b^	82.53 ± 0.50 ^a^	0.78 ± 0.06 ^b^
3	Spray drying	11.44 ± 0.14 ^b^	6.84 ± 0.64 ^c^	79.92 ± 1.17 ^ab^	1.38 ± 0.10 ^a^

Note: Values represent means of triplicate determination ± SD. Values in the same row with different superscript letters are significantly different (*p* < 0.05).

**Table 6 foods-12-02514-t006:** The color difference between different drying methods on *C. reticulata* Blanco cv. Dahongpao solid drinks.

Sample		Drying Method		
		Freeze Drying	Hot Air Drying	Spray Drying
Powder form	L*	42.00 ± 0.16 ^b^	14.93 ± 0.48 ^c^	54.87 ± 0.66 ^a^
	a*	22.83 ± 1.56 ^a^	8.07 ± 0.25 ^c^	19.6 ± 0.14 ^b^
	b*	32.57 ± 3.12 ^b^	8.23 ± 0.42 ^c^	44.40 ± 0.89 ^a^
	c*	39.77 ± 3.35 ^b^	11.5 ± 0.45 ^c^	48.53 ± 0.89 ^a^
	h°	54.87 ± 1.44 ^b^	45.63 ± 0.47 ^c^	66.33 ± 0.39 ^a^
	Color difference E	57.37 ± 2.02 ^b^	18.87 ± 0.47 ^c^	73.26 ± 0.22 ^a^
	Whiteness W	28.99 ± 2.41 ^a^	14.15 ± 0.48 ^b^	33.72 ± 1.07 ^a^
	Color difference composite index (CCI)	16.75 ± 0.82	65.74 ± 2.38	8.05 ± 0.07
Brewing Form	L*	51.73 ± 0.46 ^a^	30.23 ± 0.19 ^b^	51.47 ± 0.29 ^a^
	a*	13.33 ± 0.21 ^b^	9.30 ± 0.16 ^c^	15.97 ± 0.29 ^a^
	b*	39.97 ± 0.94 ^a^	9.60 ± 0.33 ^b^	a0.57 ± 0.40 ^a^
	c*	42.13 ± 0.97 ^a^	13.40 ± 0.33 ^b^	43.60 ± 0.43 ^a^
	h°	42.13 ± 0.97 ^c^	45.93 ± 0.39 ^b^	68.53 ± 0.17 ^a^
	Color difference E	66.72 ± 0.90 ^b^	85.01 ± 0.40 ^a^	67.45 ± 0.52 ^b^
	Whiteness W	35.92 ± 0.44 ^a^	28.96 ± 0.24 ^c^	34.76 ± 0.12 ^b^
	Color difference composite index (CCI)	6.45 ± 0.10	32.06 ± 0.37	7.65 ± 0.03

Note: Values represent means of triplicate determination ± SD. Values in the same row with different superscript letters are significantly different (*p* < 0.05).

**Table 7 foods-12-02514-t007:** Sensory evaluation of *C. reticulata* Blanco cv. Dahongpao solid drinks under different drying methods.

Project Weight	Sample Score		
	Freeze Drying	Hot Air Drying	Spray Drying
Color and luster (0.20)	55.50 ± 20.06	19.50 ± 6.50	51.50 ± 17.04
Texture (0.40)	41.50 ± 16.44	44.50 ± 14.40	38.00 ± 17.06
Smell (0.25)	57.50 ± 13.83	54.00 ± 17.00	56.00 ± 20.83
Stability (0.15)	64.00 ± 14.97	46.00 ± 16.40	57.30 ± 19.43
Total score	51.68	42.1	48.1

## Data Availability

Data is contained within the article.

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
