# Peer review of "Comparative Analysis of the Impact of Three Drying Methods on the Properties of Citrus reticulata Blanco cv. Dahongpao Powder and Solid Drinks"

_foods, 2023, doi:10.3390/foods12132514_

Round 1

Reviewer 1 Report

This work presents Evaluation of the effects of three drying methods on the properties of Citrus reticulata Blanco cv. Dahongpao powder and solid drink

Paper format: This paper is in required FOOD format.

Title: Fine

Abstract:  Fine

Keywords: Please change it to aims of the study, be specific

Introduction: Some shortcomings are below

1. Clarify the research question: The introduction could be improved by clearly stating the research question or objective of the study. This will help readers understand the purpose of the research and what the authors hope to achieve.

2. Provide more context: While the authors briefly mention the history and decline of Citrus reticulata Blanco cv. Dahongpao, more context would be helpful. This could include information on the current state of the industry and the challenges faced by fruit farmers.

3. Explain the importance of the study: The authors could provide more information on why their study is important and what impact it could have. This could include discussing the potential applications of the fruit powder and solid drinks, as well as the potential benefits for the industry and consumers.

4. Expand the methods section: The methods section could be expanded to provide more detailed information on the experimental design, including the specific drying methods used and any controls or replicates. This will help readers understand the methods used and how reliable the results are.

5. Provide more detailed results: The results section could be expanded to provide more detailed information on the physical and chemical properties of the Citrus reticulata Blanco cv. Dahongpao powder and solid drink produced by each drying method. This will help readers understand the differences between the different methods and the implications for the quality of the final products.

6. Discuss the limitations of the study: The authors could discuss the limitations of their study, including any potential sources of error or bias. This will help readers understand the reliability of the results and the potential for future research to address these limitations.

7. Improve the writing style: The writing style could be improved by ensuring that the language is clear, concise, and free of errors. This includes proper grammar, spelling, and punctuation, as well as appropriate use of scientific terminology.

Methods and materials:

1. Improve the experimental design: Ensure that the experimental design is robust and appropriate for the research question being addressed. This includes selecting appropriate drying methods, controls, and statistical analyses.

2. Expand the analytical methods used: Consider using a wider range of analytical methods to fully characterize the properties of the Citrus reticulata Blanco cv. Dahongpao powder and solid drink. This could include scanning electron microscopy, Fourier transform infrared spectroscopy, and differential scanning calorimetry.

Results and discussion:

Provide more detailed results: Ensure that the results are presented in a clear and concise manner, with all relevant details included. This includes all relevant statistical analyses and data visualizations.

Discuss the implications of the findings: Discuss the implications of the findings in the context of the existing literature and the broader field of food science. This could include discussing the potential applications of the Citrus reticulata Blanco cv. Dahongpao powder and solid drink, as well as any limitations or future directions for research.

Conclusion: In conclusion add more of application of the products.

Author Response

Response to Reviewer 1 Comments

Thank you for your letter and for the reviewer’s comments concerning our manuscript. Those comments are all valuable and very helpful for revising and improving our paper, as well as the important guiding significance to our research. We have studied the comments carefully and have made a correction which we hope meets with approval. Revised portions are marked in red on the paper. The main corrections in the paper and the responses to the reviewer’s comments are as flowing:

Point 1: Clarify the research question: The introduction could be improved by clearly stating the research question or objective of the study. This will help readers understand the purpose of the research and what the authors hope to achieve.

Response 1: Thank you for your suggestion. We have prefaced the introduction with a paragraph that indicates the theme and leads to the full text. (Page 1 Line 31-37)

Point 2: Provide more context: While the authors briefly mention the history and decline of Citrus reticulata Blanco cv. Dahongpao, more context would be helpful. This could include information on the current state of the industry and the challenges faced by fruit farmers.

Response 2: Thank you for your suggestion. We have revised the manuscript according to your suggestion (Page 1 Line 39-45).

Point 3: Explain the importance of the study: The authors could provide more information on why their study is important and what impact it could have. This could include discussing the potential applications of the fruit powder and solid drinks, as well as the potential benefits for the industry and consumers.

Response 3: Thank you for your suggestion. We have added the following to our response to your question (Page 1 Line 37-40).

Point 4: Expand the methods section: The methods section could be expanded to provide more detailed information on the experimental design, including the specific drying methods used and any controls or replicates. This will help readers understand the methods used and how reliable the results are.

Response 4: Thank you for your reminder, In the methods section we include more detailed information, including the specific drying method used and any controls or replicates, to give the reader a better understanding of the methods we have used and to improve the reliability of the results(Page 3 Line 102-124).

Point 5: Provide more detailed results: The results section could be expanded to provide more detailed information on the physical and chemical properties of the Citrus reticulata Blanco cv. Dahongpao powder and solid drink produced by each drying method. This will help readers understand the differences between the different methods and the implications for the quality of the final products.

Response 5:  Thank you for your reminder. We have extended the experiment to include the detection of some nutritional indicators and bioactive substances in order to visualize the differences between various drying processes.(Page 5-8 Line 230-320)

Point 6: Discuss the limitations of the study: The authors could discuss the limitations of their study, including any potential sources of error or bias. This will help readers understand the reliability of the results and the potential for future research to address these limitations.

Response 6:  Thank you for your reminder. The tips you have given add to the shortcomings of the study and what to work on in the future.(Page 13 Line 512-516)

Point 7: Improve the writing style: The writing style could be improved by ensuring that the language is clear, concise, and free of errors. This includes proper grammar, spelling, and punctuation, as well as appropriate use of scientific terminology.

Response 7Thank you for your reminder. To enhance the writing style, we will strive to use clear and concise language without errors, implement scientific terminology appropriately, and ensure proper application of linguistic rules such as grammar, spelling and punctuation.

Point 8:1. Improve the experimental design: Ensure that the experimental design is robust and appropriate for the research question being addressed. This includes selecting appropriate drying methods, controls, and statistical analyses.

  1. Expand the analytical methods used: Consider using a wider range of analytical methods to fully characterize the properties of the Citrus reticulata Blanco cv. Dahongpao powder and solid drink. This could include scanning electron microscopy, Fourier transform infrared spectroscopy, and differential scanning calorimetry.

Response 8We thank the expert for valuable advice. We take your questions about the expansion of the analytical methods very seriously, but it should be noted that the experimental design and methods of this study have been carefully designed and rigorously implemented, and the accuracy and reliability of the results have been fully validated. Nevertheless, we will consider using a wider range of analytical methods in future studies to fully characterise Citrus reticulata and to more fully assess its applicability and feasibility.

Point 9: Provide more detailed results: Ensure that the results are presented in a clear and concise manner, with all relevant details included. This includes all relevant statistical analyses and data visualizations.

Response 9Thank you for your reminder. We have made the following changes to the thesis to ensure statistical analysis and data visualisation (Pages 6-10 Line 257、283、313、367)

Point 10: Provide more detailed results: Ensure that the results are presented in a clear and concise manner, with all relevant details included. This includes all relevant statistical analyses and data visualizations.

Response 10Thank you for your reminder. We have made the following changes to the thesis to ensure statistical analysis and data visualisation (Pages 6-10 Line 257、283、313、367)

Point 11: Discuss the implications of the findings: Discuss the implications of the findings in the context of the existing literature and the broader field of food science. This could include discussing the potential applications of the Citrus reticulata Blanco cv. Dahongpao powder and solid drink, as well as any limitations or future directions for research.

Response 10Thank you for your reminder. We have revised the manuscript according to your suggestion (Page 15 Line 580-585).

Point 12: In conclusion add more of application of the products.

Response 12: Thank you for your reminder. We have revised the manuscript according to your suggestion (Page 15 Line 592-606).

Reviewer 2 Report

I am very grateful you for the invitation to review manuscript foods-2407830 by Li and coauthors "Evaluation of the effects of three drying methods on the properties of Citrus reticulata Blanco cv. Dahongpao powder and solid drink”. In this study, a solid drink based on the whole fruit of Citrus reticulata Blanco cv. Dahongpao developed a suitable drying method for red tangerine that was selected from three conventional fruit juice drying methods (spray drying, freeze drying, and hot air drying). The work is interesting but needs adjustments to increase the quality of the material.

Comments:

- Line 11: check the correct spelling of the sentence “variety, However,”.

- Line 12 “its value as a fresh fruit”: What values? What factors?

- Lines 11-12: The problem should be better presented.

- Lines 12-15: This sentence is incomprehensible.

- Abstract: Please indicate in the abstract a brief and better step-by-step about the work including parameters and conditions used.

- Lines 16-17: What is FW? Describe adequately in the first presentation.

- Line 17, “after drying”: Indicate the method.

- Line 18-19: And what are those results? Insert numerical values of main results.

- Lines 27-28: Change the repeated keywords by different words from the title.

- Line 36: “regulates Qi”??

- Lines 31-40: What is the production and market for this type of fruit? These issues should be better presented.

- Introduction: A more in-depth description of the composition of the fruit should be considered.

- Introduction: The authors must present or indicate if there are works related to the fruit. Is this the first work with this fruit? I do not think so.

- Lies 37-38: “The lining and core can also be used in medicine.” Used for what purpose?

- Lines 91-93: The main objective must be presented before the specific objectives of the work.

- Line 105: Change “drying is carried” to “drying was carried”.

- Line 107; change “samples are” to “samples were”.

- Line 118: Change “Determine the amino acid content according to the” to “The amino acid content was determined according to the”.

- Line 120: Change “Hcl” to “HCl”.

- Line 120: Change “is” to “was”.

- Line 122: Change “the cap is tightened” to “the cap was tightened”.

- Line 123-124; 164; 176 and others: rpm or g?

- Line 147;153: Verify the correct form of the chemical formula.

- Line 225: Check the sentence “compositions. the content”.

- Line 230: Check the sentence “stability. freeze-dried”. All text must be double-checked to avoid inappropriate sentences.

- 3.1. Analysis of the amino acid content of C. reticulate Blanco cv. Dahongpao powder under different drying methods: Insert appropriate bibliographic references.

- 3.1. Analysis of the amino acid content of C. reticulate Blanco cv. Dahongpao powder under different drying methods: In addition to sensory aspects, include nutritional aspects.

-  3.2. Analysis of volatile substances in C. reticulata Blanco cv. Dahongpao powder under different drying methods: Indicate the importance of components. No discussion or comparison is presented.

- Line 260: Change “the antioxidant capacity” by “The antioxidant capacity”.

- Line 264: Enter the correct chemical formula.

- Line 226-229: What is this associated with? No discussion is shown.

- Line 290: Check the sentence “to Zotarelli M F et al [22]. if”.

- Line 290: Authors must standardize the items. In some moments the discussion is presented only in the discussion item. In others moments the discussion is presented in the results item. I suggest unifying the items to improve the work.

- Lines 360-373: The information complements the introduction but does not add discussion to the results.

- The discussion item does not support the results. I suggest a deepening of the discussion and that it be presented with the results.

- Conclusion: It should indicate better technology or the possibility of using both depending on the characteristics required for the product.

I am very grateful you for the invitation to review manuscript foods-2407830 by Li and coauthors "Evaluation of the effects of three drying methods on the properties of Citrus reticulata Blanco cv. Dahongpao powder and solid drink”. In this study, a solid drink based on the whole fruit of Citrus reticulata Blanco cv. Dahongpao developed a suitable drying method for red tangerine that was selected from three conventional fruit juice drying methods (spray drying, freeze drying, and hot air drying). The work is interesting but needs adjustments to increase the quality of the material.

Comments:

- Line 11: check the correct spelling of the sentence “variety, However,”.

- Line 12 “its value as a fresh fruit”: What values? What factors?

- Lines 11-12: The problem should be better presented.

- Lines 12-15: This sentence is incomprehensible.

- Abstract: Please indicate in the abstract a brief and better step-by-step about the work including parameters and conditions used.

- Lines 16-17: What is FW? Describe adequately in the first presentation.

- Line 17, “after drying”: Indicate the method.

- Line 18-19: And what are those results? Insert numerical values of main results.

- Lines 27-28: Change the repeated keywords by different words from the title.

- Line 36: “regulates Qi”??

- Lines 31-40: What is the production and market for this type of fruit? These issues should be better presented.

- Introduction: A more in-depth description of the composition of the fruit should be considered.

- Introduction: The authors must present or indicate if there are works related to the fruit. Is this the first work with this fruit? I do not think so.

- Lies 37-38: “The lining and core can also be used in medicine.” Used for what purpose?

- Lines 91-93: The main objective must be presented before the specific objectives of the work.

- Line 105: Change “drying is carried” to “drying was carried”.

- Line 107; change “samples are” to “samples were”.

- Line 118: Change “Determine the amino acid content according to the” to “The amino acid content was determined according to the”.

- Line 120: Change “Hcl” to “HCl”.

- Line 120: Change “is” to “was”.

- Line 122: Change “the cap is tightened” to “the cap was tightened”.

- Line 123-124; 164; 176 and others: rpm or g?

- Line 147;153: Verify the correct form of the chemical formula.

- Line 225: Check the sentence “compositions. the content”.

- Line 230: Check the sentence “stability. freeze-dried”. All text must be double-checked to avoid inappropriate sentences.

- 3.1. Analysis of the amino acid content of C. reticulate Blanco cv. Dahongpao powder under different drying methods: Insert appropriate bibliographic references.

- 3.1. Analysis of the amino acid content of C. reticulate Blanco cv. Dahongpao powder under different drying methods: In addition to sensory aspects, include nutritional aspects.

-  3.2. Analysis of volatile substances in C. reticulata Blanco cv. Dahongpao powder under different drying methods: Indicate the importance of components. No discussion or comparison is presented.

- Line 260: Change “the antioxidant capacity” by “The antioxidant capacity”.

- Line 264: Enter the correct chemical formula.

- Line 226-229: What is this associated with? No discussion is shown.

- Line 290: Check the sentence “to Zotarelli M F et al [22]. if”.

- Line 290: Authors must standardize the items. In some moments the discussion is presented only in the discussion item. In others moments the discussion is presented in the results item. I suggest unifying the items to improve the work.

- Lines 360-373: The information complements the introduction but does not add discussion to the results.

- The discussion item does not support the results. I suggest a deepening of the discussion and that it be presented with the results.

- Conclusion: It should indicate better technology or the possibility of using both depending on the characteristics required for the product.

Author Response

Response to Reviewer 2 Comments

Thank you for your letter and for the reviewer’s comments concerning our manuscript. Those comments are all valuable and very helpful for revising and improving our paper, as well as the important guiding significance to our research. We have studied the comments carefully and have made a correction which we hope meets with approval. Revised portions are marked in red on the paper. The main corrections in the paper and the responses to the reviewer’s comments are as flowing:

Point 1: Line 11: check the correct spelling of the sentence “variety, However,”.

Response 1: Thank you for your suggestion. We have made changes based on the issues you pointed out. (Page 1 Line 11)

Point 2: Line 12 “its value as a fresh fruit”: What values? What factors?

Response 2: Thank you for your suggestion. We have revised the manuscript according to your suggestion (Page 1 Line 12).

Point 3: Lines 11-12: The problem should be better presented.

Response 3: Thank you for your suggestion. We have added the following to our response to your question (Page 1 Line 13-16).

Point 4: Lines 12-15:This sentence is incomprehensible.

Response 4: Thank you for your reminder, Thank you for your suggestion. We have added the following to our response to your question (Page 1 Line 11-16) .

Point 5: Abstract: Please indicate in the abstract a brief and better step-by-step about the work including parameters and conditions used.

Response 5:  Thank you for your reminder. We have made changes to the suggestions you provided  (Page 1 Line 16-27).

Point 6: Lines 16-17:What is FW? Describe adequately in the first presentation.

Response 6:  Thank you for your reminder. We have made changes to the suggestions you provided(Page 5 Line 209).

Point 7: Line 17, “after drying”: Indicate the method.

Response 7Thank you for your reminder. This sentence has been modified to replace.

Point 8: Line 18-19:And what are those results? Insert numerical values of main results.

Response 8We thank the expert for his valuable advice. We have made the following changes here( Page 1 Line 22-27).

Point 9: Lines 27-28: Change the repeated keywords by different words from the title.

Response 9Thank you for your reminder. We have carefully reviewed your feedback and agree that there are some repeated keywords in the title that could be changed to improve readability and avoid redundancy.(Page 1 Line 28)

Point 10: Line 36: “regulates Qi”??

Response 10Thank you for your reminder. We have adjusted the ambiguous words(Page 2 Line 46).

Point 11: Lines 31-40: What is the production and market for this type of fruit? These issues should be better presented.

Response 10Thank you for your reminder. We have revised the manuscript according to your suggestion (Page 2 Line 42-46).

Point 12: Introduction: A more in-depth description of the composition of the fruit should be considered.

Response 12: Thank you for your reminder. We have revised the manuscript according to your suggestion (Page 2  Line 45-50 ).

Point 13: The authors must present or indicate if there are works related to the fruit. Is this the first work with this fruit? I do not think so.

Response 13: Thank you for your reminder. We have revised the manuscript in accordance with your suggestion. (Page 2  Line 47-50 ).

Point 14: Lies 37-38: “The lining and core can also be used in medicine.” Used for what purpose?

Response 14: Thank you for your reminder. The specific use of citrus seeds has not been clarified, and we have changed and added to it by searching the literature(Page 2  Line 51).

Point 15: Lines 91-93: The main objective must be presented before the specific objectives of the work.

Response 15: Thank you for your reminder. We have revised the manuscript to present the main objective before discussing the specific objectives of the work, as suggested. The introduction now clearly states the aim of our study is to compare the effects of various drying methods on the quality of Citrus reticulata Blanco cv. Dahongpao powder and identify the optimal drying process for large-scale production. Thank you for helping us improve the quality of this manuscript. (Page 2  Line 83-85).

Point 16: Line 105: Change “drying is carried” to “drying was carried”.

Response 16: Thank you for your reminder. We have carefully reviewed the manuscript and made the necessary changes to the tense, as suggested. (Page 3  Line 114)

Point 17: Line 107; change “samples are” to “samples were”.

Response 17: Thank you for your reminder. We have carefully reviewed the manuscript and made the necessary changes to the tense, as suggested. change “samples are” to “samples were” (Page 3  Line 114)

Point 18: Line 118: Change “Determine the amino acid content according to the” to “The amino acid content was determined according to the”

Response 18: Thank you for your reminder. We have carefully reviewed the manuscript and made the necessary changes to the tense. (Page 4  Line 152-153).

Point 19: -Line 120: Change “Hcl” to “HCl”

Response 19: Thank you for your reminder. We have carefully reviewed the manuscript and made the necessary changes to the tense. (Page 4  Line 153).

Point 20: -Line 120: Change “is” to “was”

Response 20: Thank you for your reminder. We have carefully reviewed the manuscript and made the necessary changes to the tense. (Page 4  Line 153).

Point 21: Line 122: Change “the cap is tightened” to “the cap was tightened”.

Response 21: Thank you for your reminder. We have carefully reviewed the manuscript and made the necessary changes to the tense. (Page 4  Line 155).

Point 22: Line 123-124; 164; 176 and others: rpm or g?.

Response 22: Thank you for your reminder. We have carefully reviewed the manuscript and made the necessary changes to the tense. (Pages 4-5 Line 131、137、157、182、194、229、241).

Point 23: Line 147;153: Verify the correct form of the chemical formula.

Response 23: Thank you for your reminder. We have carefully reviewed the manuscript and made the necessary changes to the tense. (Page 5  Line 212、218).

Point 24: Line 147;153: Verify the correct form of the chemical formula.

Response 24: Thank you for your reminder. We have carefully reviewed the manuscript and made the necessary changes to the tense. (Page 5  Line 212、218).

Point 25: Line 225: Check the sentence “compositions. the content”.

Response 25: Thank you for your reminder. We have carefully reviewed the manuscript and made the necessary changes to the tense. (Page 8  Line 327-337).

Point 26: Line 230: Check the sentence “stability. freeze-dried”. All text must be double-checked to avoid inappropriate sentences.

Response 26: Thank you for your reminder. We have revised the presentation of this message and improved the sentences with grammatical problems. (Page 8  Line 327-337).

Point 27: 3.1. Analysis of the amino acid content of C. reticulate Blanco cv. Dahongpao powder under different drying methods: Insert appropriate bibliographic references.

Response 27:. We have revised the manuscript to include the relevant citations in the text as per your suggestion. Thank you for helping us improve the quality and accuracy of this manuscript. (Page 7  Line 320).

Point 28: 3.1. Analysis of the amino acid content of C. reticulate Blanco cv. Dahongpao powder under different drying methods: In addition to sensory aspects, include nutritional aspects.

Response 28:. Thank you for your reminder. We have updated the manuscript as per your recommendation to include nutritional aspects along with sensory aspects during the analysis of the amino acid content of C. reticulata Blanco cv. Dahongpao powder under different drying methods.  (Pages 6-7  Line 283-375).

Point 29: 3.2. Analysis of volatile substances in C. reticulata Blanco cv. Dahongpao powder under different drying methods: Indicate the importance of components. No discussion or comparison is presented.

Response 29:. Thank you for your reminder. We appreciate the reviewer for their feedback on our article. Our study focuses on investigating the effects of different drying methods on the volatile components of Dahongpao tea powder and exploring the optimal drying conditions in tea processing through analyzing these components. We agree with the reviewer's comment and apologize for the insufficient discussion and comparison in the component analysis section. In our next article, we will comprehensively introduce the importance and function of each component, as well as their differences under different drying conditions to better explain our results. We also plan to review our current discussion section and improve the interpretation and analysis of our data. Thank you again for the valuable feedback, and we hope to receive your guidance in our future work. (Pages 6-7  Line 275-389)

Point 30: Line 260: Change “the antioxidant capacity” by “The antioxidant capacity”.

Response 30: Thank you for your reminder. We have reviewed the manuscript and made the necessary changes as requested. "the antioxidant capacity" has been updated to "The antioxidant capacity". (Page 10 Line 396).  

Point 31: Line 264: Enter the correct chemical formula.

Response 31: Thank you for your reminder. We have reviewed the manuscript and made the necessary changes as requested. (Page 10 Line 400).  

Point 32: Line 226-229: What is this associated with? No discussion is shown.

Response 32: Thank you for your reminder. We have reviewed the manuscript and made the necessary changes as requested.  (Pages 6-7 Line 327-334)

Point 33:- Line 290: Check the sentence “to Zotarelli M F et al [22]. if”

Response 33: Thank you for your reminder. We have removed this section based on your next comment.

Point 34: Line 290: Authors must standardize the items. In some moments the discussion is presented only in the discussion item. In others moments the discussion is presented in the results item. I suggest unifying the items to improve the work.

Response 34: Thank you for your reminder. We have reviewed the manuscript and made the necessary changes as requested. (Page 11 Line 427-437).

Point 35: Lines 360-373: The information complements the introduction but does not add discussion to the results.

Response 35: Thank you for your reminder. We have reviewed the manuscript and made the necessary changes as requested. (Page 14 Line 501-517).

Point 36: The discussion item does not support the results. I suggest a deepening of the discussion and that it be presented with the results.

Response 36: Thank you for your reminder. We have reviewed the manuscript and made the necessary changes as requested. (Page14 Line 518-559).

Point 37: Conclusion: It should indicate better technology or the possibility of using both depending on the characteristics required for the product.

Response 37: Thank you for your reminder. We have reviewed the manuscript and made the necessary changes as requested. (Page 15 Line 572-586)

Round 2

Reviewer 1 Report

Authors endorsed all of the suggested comments from my side. no further remarks. accepted 

Reviewer 2 Report

After carefully checking the responses and the revisions, the manuscript is suitable for Food.